# Mechanisms of Action of Non-Canonical ECF Sigma Factors

**DOI:** 10.3390/ijms23073601

**Published:** 2022-03-25

**Authors:** Francisco Javier Marcos-Torres, Aurelio Moraleda-Muñoz, Francisco Javier Contreras-Moreno, José Muñoz-Dorado, Juana Pérez

**Affiliations:** 1Estación Experimental del Zaidín, Consejo Superior de Investigaciones Científicas, 18011 Granada, Spain; fjmarcos@ugr.es; 2Departamento de Microbiología, Facultad de Ciencias, Universidad de Granada, Avda, Fuentenueva s/n, 18071 Granada, Spain; currocm@ugr.es (F.J.C.-M.); jdorado@ugr.es (J.M.-D.)

**Keywords:** ECF sigma factor, transcriptional regulation, stress response

## Abstract

Extracytoplasmic function (ECF) sigma factors are subunits of the RNA polymerase specialized in activating the transcription of a subset of genes responding to a specific environmental condition. The signal-transduction pathways where they participate can be activated by diverse mechanisms. The most common mechanism involves the action of a membrane-bound anti-sigma factor, which sequesters the ECF sigma factor, and releases it after the stimulus is sensed. However, despite most of these systems following this canonical regulation, there are many ECF sigma factors exhibiting a non-canonical regulatory mechanism. In this review, we aim to provide an updated and comprehensive view of the different activation mechanisms known for non-canonical ECF sigma factors, detailing their inclusion to the different phylogenetic groups and describing the mechanisms of regulation of some of their representative members such as EcfG from *Rhodobacter sphaeroides*, showing a partner-switch mechanism; EcfP from *Vibrio parahaemolyticus*, with a phosphorylation-dependent mechanism; or CorE from *Myxococcus xanthus*, regulated by a metal-sensing C-terminal extension.

## 1. Introduction

In order to survive to the ever-changing environmental conditions, bacteria have developed a set of diverse regulatory mechanisms to sense and respond to external and internal signals. These signal-transduction mechanisms are generally classified in four main groups, also known as the four pillars of signal transduction: one- and two-component systems, extracytoplasmic function (ECF) sigma factors, and Ser/Thr protein kinases (STPK) [1,2]. Sigma factors are the subunits of the RNA polymerase (RNAP) involved in promoter recognition and initiation of the transcription process [3,4]. ECF sigma factors represent a specific group of these subunits (Group 4 of sigma factors), harboring only two of the four conserved domains of sigma factors (σ2 and σ4) [3].

ECF sigma factors are specialized in the response to specific conditions, such as environmental stress, differentiation, or life cycle stage. Typically, in the absence of the stimulus, ECF sigma factors are kept sequestered by their co-transcribed anti-sigma factors. Upon the arrival of the stimulus, these membrane-anchored anti-sigma factors act as the sensor part of these signal-transduction mechanisms and release the sigma factor, normally through the regulated intramembrane proteolysis (RIP) of the anti-sigma factor, to start the transcription of genes in response to the triggering stimulus. Even though this would represent the most common and widespread mode of regulation for canonical ECF sigma factors, a broad diversity of regulatory mechanisms has been described for these proteins since their discovery [5,6,7]. This diversity was first evident after the phylogenetic analysis performed by Staroń et al. in 2009, which resulted in the establishment of 66 ECF sigma factor groups with very distinctive features [1]. This classification, plus the later additions summing up to more than 94 groups, was used to define the different mechanisms of activation of ECF sigma factors [5,6]. However, since the last attempt to do such a regulatory classification by Pinto and Mascher in 2016, new discoveries regarding ECF sigma factors have been published, including the most recent phylogenetic analysis performed by Casas-Pastor et al. in 2021, which resulted in the formation of 157 new ECF groups, with the re-definition, disappearance, or expansion of many of the original groups [6,7]. Many defined ECF groups contain only a single sigma member. This makes the sigma grouping problematic and shows a wide variety of sigma types. In this mini-review, we aim to provide an updated general and comprehensive view of the complexity of regulatory mechanisms of the non-canonical ECF sigma factors.

What we define here as canonical ECF sigma factors are those following the general mechanism of regulation of the majority of ECF sigma factors. These canonical regulators are under control of their cognate membrane-bound anti-sigma factors, transferring an extracytoplasmic signal across the bacterial membranes to trigger a genetic response, regardless of the use of additional regulatory elements as in the case of cell-surface signaling systems or the anti-anti-sigma factors. Thus, the non-canonical ECF sigma factors would be those whose regulatory mechanisms involving a cytoplasmic signal, or where the signal transduction across membranes involves the action of additional mechanisms, such as two-component systems. According to this definition, we can distinguish between two groups of non-canonical ECF sigma factors: (1) those under the control of a soluble anti-sigma factor, and (2) those not associated to an anti-sigma factor.

## 2. ECF Sigma Factors with a Soluble Anti-Sigma Factor

Even though regulatory soluble anti-sigma factors normally share little sequence homology with their membrane-bound functional homologues, they do share a striking structural homology leading to the characterization of the anti-sigma domain present in most of these proteins [8]. This diversity of anti-sigma factors makes the identification of some of their most divergent members a complicated task, resulting in many proteins having a putative anti-sigma factor status.

Three mechanisms of regulation have been reported for ECF sigma factors that function with a soluble anti-sigma factor (Table 1): sigma factors regulated by conformational change, sigma factors regulated by partner switch, and sigma factors regulated by a mechanosensing complex.

Other less-understood ECF sigma factors with soluble anti-sigma factors encompass those included in groups ECF125 and ECF127, which are encoded next to a gene with high similarity with the anti-sigma factors of group ECF121; those included in groups ECF270 and ECF271, which are regulated by proteins with a Zn-dependent anti-sigma domain (ZAS); and those included in groups ECF286 and ECF292, which are encoded next to soluble proteins from the Asp23 family, predicted to act as anti-sigma factors [7]. However, due to the lack of experimental data to confirm their activation mechanisms, these ECF sigma factors will not be discussed in this review.

### 2.1. ECF Sigma Factors Regulated by Conformational Change

These ECF sigma factors represent the closest activation mechanism to the canonical one. However, contrary to the membrane-bound anti-sigma factors, where the location of the sensor domain enables them to respond to signals present in the periplasm or the outer membrane, these soluble proteins generally respond to cytoplasmic signals. These soluble anti-sigma factors brandish a ZAS domain, which normally responds to different types of oxidative stress. Examples of this mechanism would include RpoE from *Rhodobacter sphaeroides* (ECF11), and SigH and SigE from *Mycobacterium tuberculosis* (ECF12 and ECF14, respectively) [9,10,11].

RpoE is the master regulator of the singlet oxygen and organoperoxides oxidative stress response in *R. sphaeroides*. Under normal conditions, RpoE is kept inactive by the soluble protein ChrR, which interacts with the regions binding to the RNAP and the DNA, acting as an anti-sigma factor. ChrR regulators have two well-defined domains: an N-terminal ZAS domain, and a C-terminal cupin-like domain (CLD). Whereas the first one is characterized by the two conserved cysteines that coordinate the binding to Zn^2+^ to sequester the ECF sigma factor, the last one would act as the sensor region of this protein, also binding to a Zn^2+^ atom [9]. Even though in many oxidative stress sensing ZAS proteins, the dissociation between sigma and anti-sigma proteins takes place by the conformational change derived from the oxidative damage of the Zn^2+^ ligands [15], it has been shown that singlet oxygen stimulates ChrR proteolysis [16]. However, it is unclear whether this turnover plays a key role in the release of RpoE or how this regulated proteolysis may occur [9,17].

The oxidative-stress sigma factors SigH and SigE from *M. tuberculosis* are regulated by the soluble proteins RshA and RseA, respectively (Figure 1A). These regulatory proteins have a ZAS domain that, as mentioned above, coordinates a Zn^2+^ atom to bind to their respective ECF sigma factors and keep them inactive. Upon oxidative stress conditions, the Zn^2+^ ligand is released from both proteins, causing a conformational change that disrupts their binding to their ECF sigma factors [10,11,18]. Both ECF sigma factors have an additional level of regulation mediated by the Ser/Thr protein kinase PknB, since these sigma factors are also released upon phosphorylation of the anti-sigma factor [18,19,20].

### 2.2. ECF Sigma Factors Regulated by Partner Switch

ECF sigma factors regulated by partner switch, also known as sigma factor mimicry, are normally kept inactive by a soluble protein acting as an anti-sigma factor. However, in this case, the signaling mechanism additionally involves the action of a two-component system, consisting of a histidine kinase and a response regulator. Contrary to other systems, here the anti-sigma factor does not have a sensory role, but simply prevents the ECF sigma factor from binding to their promoter sequences. The stimulus is instead detected by the histidine kinase which, in turn, will phosphorylate its cognate response regulator. The response regulator has a region with a high structural similarity to the ECF sigma factor itself and, once phosphorylated, binds to the anti-sigma factor more efficiently than to the ECF sigma factor. The best understood ECF sigma factor regulated by partner switch is EcfG from *Methylobacterium extorquens* (ECF15). EcfG is the central regulator of the general stress response in Alphaproteobacteria [13]. In the absence of stress, the anti-sigma factor NepR binds to the sigma factor EcfG and keeps it inactive. Upon stress conditions, the response regulator PhyR gets phosphorylated by its cognate histidine kinase and mimics the sigma factor EcfG, replacing it on the NepR binding site, and releasing the active EcfG (Figure 1B). This mechanism, rather than as a paired system, is proposed to work as a ternary complex, where NepR would be required for PhyR correct phosphorylation and subsequent conformational change [13]. This complex formation seems to be mediated by the intrinsically disordered region of the anti-sigma factor NepR, which may become structured to interact with PhyR [21].

### 2.3. ECF Sigma Factors Regulated by a Mechanosensing Complex

This mechanism of regulation is one of the latest additions to this classification, and counts with only one studied case: SigX from *Pseudomonas aeruginosa* (ECF102). The putative anti-sigma factor is a soluble small protein named CfrX. This protein has been proposed to be part of a mechanosensing complex together with the outer membrane porin OprF, and the ion channel CmpX that coordinates activation of SigX (Figure 1C). SigX is involved in the response to membrane stress and cold-shock, and regulates a plethora of genes involved in motility, iron uptake, cell wall integrity, fatty acids biosynthesis, and virulence [14,22,23,24].

## 3. ECF Sigma Factors Not Associated with an Anti-Sigma Factor

This group involves the most diverse and perhaps the most interesting ECF sigma factors from an evolutionary and biotechnological point of view. The independence of many of these regulators from additional proteins bypasses the limitations of their use for heterologous expression and, in some cases, can reflect many of the gene fusion events happening throughout bacterial evolution [25]. The many regulatory mechanisms known for these ECF sigma factors include transcriptional regulation, conformational changes, proteolysis, phosphorylation, and the use of N- and C-terminal regulatory extensions (Table 2).

Besides the already well-defined regulatory mechanisms, there are many groups of ECF sigma factors without cognate anti-sigma factors that are potentially regulated by a non-canonical mechanism yet to be characterized. Among the conserved proteins in their genetic neighborhood that could be involved in their regulatory mechanisms are found proteins with a DUF3470 and an iron-sulfur binding domain (ECF58), 6-O-methylguanine DNA methyltransferases (ECF122), glycosyltransferases (ECF248), ABC transporters and AAA ATPases (ECF257), and PadR transcriptional repressors (ECF265) [7].

### 3.1. ECF Sigma Factors Transcriptionally Regulated

These ECF sigma factors are expressed under control of another signal-transduction mechanism, which directly activates the transcription of the active ECF sigma factor in a cascade fashion.

One of the best known ECF sigma factors regulated by this mechanism is HrpL from *Pseudomonas syringae* (ECF32). The expression of this regulator is under control of the enhancer-binding protein complex HrpRS, which activates transcription of the *hrpL* gene from a sigma-54 dependent promoter [36]. HrpL is involved in the regulation of most type 3 secretion system (T3SS) genes, which are required for plant infection by this pathogen. The regulatory cascade controlling expression of *hrpL* can expand beyond HrpRS, involving more than 20 factors—such as the two-component systems RhpRS and CorRS, the one-component system AefR, or the protease LonD—and can respond to environmental cues such as changes in nutrient levels, temperature, or osmotic pressure [26,36]. The main negative regulation of this mechanism is performed by HrpL itself in a negative feedback fashion, where the complex formed by the RNAP with HrpL blocks the transcription from the sigma-54 dependent promoter [37].

Other well-known examples would include SigE and SigQ from *Streptomyces coelicolor* (ECF39). SigE is under control of the two-component system CseBC and the lipoprotein CseA, which respond to cell-wall damage caused by a diverse set of stress factors such as lysozyme or antibiotics targeting the peptidoglycan, such as ampicillin or vancomycin (Figure 2A) [27,38]. SigQ, on the other hand, is under control of the two-component system AfsQ1/Q2, and regulates sporulation and the synthesis of many antibiotics, probably in response to alterations in the nitrogen metabolism or the C/N/P ratio [39,40].

Besides these known examples, several ECF sigma factors have been proposed to be transcriptionally regulated. Thus, the sigma factor PA3285 (ECF293) from *P. aeruginosa* is predicted to be regulated by the iron-responsive ECF sigma factor PvdS [14,23,41]. Similarly, members of the groups ECF203 and ECF234 are possibly regulated by TetR repressors (in the case of group ECF203) or two-component systems (in the case of group ECF234) conserved in their genetic neighborhoods [7]. SigH from *Porphyromonas gingivalis* (ECF114) is induced in the presence of oxygen in a SigH-independent manner, suggesting that it is also activated by transcriptional regulation [28].

### 3.2. ECF Sigma Factors Regulated by Conformational Changes

This mechanism of action has been proposed for SigC from *M. tuberculosis* (ECF36), although further experimental data will be required to be fully established (Figure 2B). This sigma factor is involved in virulence during infection and in copper acquisition under metal-limiting conditions [42,43]. Despite being highly expressed during most of the *M. tuberculosis* life cycle, SigC is not often found in complex with the RNAP core enzyme, suggesting that the protein might be normally translated in an inactive or unstable conformation [44]. The fact that in vitro analyses showed that both domains of the protein are able to interact, occluding the DNA binding sites of the sigma factor, suggests that there is a conformational change required for SigC to be active [45].

### 3.3. ECF Sigma Factors Regulated by Proteolysis

This mechanism of regulation, mainly found in Actinobacteria, uses different peptidases—such as Clp proteases, subtilases, or carboxipeptidases—to keep the system inactive by proteolytic degradation of the sigma factor.

To date, only AntA (ECF282) from *Streptomyces albus* has been shown experimentally to exhibit this kind of proteolysis-regulated mechanism. The protein AntA regulates part of the gene cluster involved in the synthesis of the bioactive compound antimycin. This regulator holds a C-terminal AA motif (a di-alanine at the C terminus) which acts as a direct target for the protease ClpXP to degrade the protein (Figure 2C), preventing its accumulation and transcription of the genes under its control [29,46]. Even though this sigma factor is transcriptionally regulated by the LuxR repressor FscRI, in vivo studies have shown that is the action of the ClpXP protease which plays a major role on the conditional presence of AntA during different stages of the life cycle of *S. albus* [29].

The use of proteolytic enzymes to control the activity of ECF sigma factors in the absence of an anti-sigma factor has also been proposed for the members of the group ECF54 due to the conserved presence of proteins holding carboxypeptidase, subtilase, and caspase HetF associated with Tprs (CHAT) domains in their genetic context [7,47].

### 3.4. ECF Sigma Factors Regulated by Phosphorylation

A significant number of ECF sigma factors have been predicted to be regulated by phosphorylation, an activation mechanism very different from that of canonical ECFs. This prediction was based on microsynteny studies, which revealed that they are encoded in the proximity of a gene for an STPK [1,5,7].

One of the ECF sigma factors regulated by phosphorylation is EcfK from *Xanthomonas citri* [48]. This sigma factor (included in group ECF43) is encoded next to the STPK PknS, in a region that encodes a type VI secretion system (T6SS) required to protect cells from predation by the amoeba *Dictyostelium discoideum*. In this study, it has been demonstrated that PknS is required to induce the expression of the T6SS, and that a phosphomimetic mutation in Thr51 (the residue predicted to be phosphorylated) by a glutamic acid is able to upregulate the expression of the T6SS in a Δ*pknS* mutant. According to these data, it has been postulated that PknS, in the presence of *D. discoideum*, activates EcfK by phosphorylation, which upregulates the T6SS to resist predation [48].

More recently, it has been demonstrated that EcfP from *Vibrio parahaemolyticus* (ECF43) is activated by phosphorylation by the STPK PknT in response to polymyxin to regulate the expression of a regulon that confers resistance to this antibiotic [30]. Contrary to canonical ECF sigma factors, EcfP is intrinsically inactive (Figure 2D). This inactivity relies on the fact that it lacks a DAED motif in the σ2.2 region (this name refers to the amino acids found in this motif), which contains the negatively charged residues usually required to interact with positively charged residues of the β’ subunit of the RNAP [49,50,51,52]. In contrast, EcfP shows in this region an STTA motif (also referred to the residues found in the same positon), in which the second Thr is the residue phosphorylated by PknT [30]. This phosphorylation provides the negative charge required for interaction with the core RNAP. In this signal-transduction pathway, it is hypothesized that PknT somehow senses the stress originated by polymyxin, inducing the kinase activity and the phosphorylation of EcfP to express genes involved in polymyxin resistance.

As mentioned above, *V. parahaemolyticus* EcfP is included in group ECF43 of sigma factors, and all members of this group lack the DAED motif in the σ2.2 region, indicating that all of them function in a similar manner, being activated by phosphorylation by an STPK encoded in the proximity of the sigma factor gene [7]. In fact, Iyer et al. (2020) also demonstrated that another ECF43 sigma factor from *Hyphomonas neptunium* is activated in a similar manner by an STPK [30]. According to the most recent classification of ECF sigma factors, a total of six groups (including ECF43) have been proposed to be activated by phosphorylation, because they are not usually co-expressed with an anti-sigma factor, but with an STPK [7]. However, ECFs from groups different from ECF43 may function in a different manner as they exhibit a DAED motif (or a very similar sequence with negatively charged residues) to interact with the RNAP. Nevertheless, bioinformatic analyses have shown that either Ser or Thr residues appear in some groups in or around this motif that could be the target of an STPK [30]. Characterization of more ECF sigma factors of different groups postulated to be regulated by phosphorylation will be necessary to elucidate their mechanisms of action, which may differ from that reported for EcfP.

### 3.5. ECF Sigma Factors with Regulatory Extensions

These ECF sigma factors, while lacking a regulatory anti-sigma factor, conserve a N-terminal or a C-terminal extension that typically takes over the sensory and/or regulatory role of those missing proteins. In some cases, these regulatory extensions seem indeed to have been originated by an ancient translational fusion with their anti-sigma factor. ECF sigma factors with regulatory C-terminal extensions were first described in *Myxococcus xanthus* [32] as well as in *Bacillus licheniformis* and *R. sphaeroides* [53]. After that, many other ECF sigma factors with regulatory extensions have been identified bioinformatically [6,7,25] or experimentally demonstrated [31,33,54]. The latest classification of ECF sigma factors shows that members of at least 26 ECF phylogenetic groups present N- or C-terminal extensions predicted to have a key regulatory role [7]. To date, three main molecular mechanisms governing this regulation have been described: activated by conformational change, activated by protein interaction, and activated by proteolysis. These mechanisms comprehend 11 different ECF sigma factor groups, while the other 15 remain to be studied and may show additional regulatory mechanisms (Table 2).

#### 3.5.1. Activated by Conformational Change

The C-terminal extension of these proteins is a structural domain predicted to interact with the σ2 and σ4 domains of the ECF sigma factor. This extension, when binding to specific subtracts (likely small molecules), modulates the conformation of the ECF sigma factor, allowing promoter recognition, and permitting the ECF to carry out its activity. The ECFs following this regulation have been recently assigned to groups ECF41 and ECF238.

CorE and CorE2 from *M. xanthus* (ECF238) represent the best understood group of ECF sigma factors with C-terminal extensions. These proteins hold a cysteine-rich domain (CRD) in the C-terminus and a CXC motif in the linker between σ2 and σ4, both essential for activity (Figure 3A) [2,32,33]. The re-classification of CorE-like sigma factors into the new ECF238 group raises some challenges compared to their previous classification into the former ECF44 group. First of all, CorE—the founding member of these regulators—is now unclassified under the new classification criteria. Second, several of the signature features of the CorE-like sigma factors, like the presence of the CXC motif, are only conserved for the members of the former ECF44 group, but not for the remaining ~85% members of the new ECF238 group. The difficulties of accurately assigning CorE-like sigma factors into this group (like in the case of CorE), together with the lack of uniformity within the new ECF238 group, argues for a careful re-examination of this group and probably for a restitution of the former ECF44 phylogenetic group.

Despite their similarities, both characterized CorE-like proteins exhibit clear differences. CorE, which is involved in copper homeostasis, undergoes an activation/inactivation process dependent on the copper redox state, in which CorE is activated by Cu^2+^, while it is quickly inactivated by Cu^+^ due to the strong reducing conditions of the cytoplasm [32,55]. However, CorE2 responds to Cd^2+^ and Zn^2+^, and consequently, the inactivation is observed in the absence of these metals. The Cys distribution of their C-terminal extensions has been demonstrated to be responsible for their metal specificity and the type of response [32,33]. Due to the abundance of cysteine residues in their C-terminal extensions, other ECF sigma factors—such as the members of groups ECF287 and ECF288—have been suggested to have a similar regulatory mechanism to the CorE-like sigma factors.

On the other hand, ECF sigma factors from the ECF41 group, exemplified by SigJ and SigI from *M. tuberculosis*, as well as Ecf41Bli from *B. licheniformis* and *R. sphaeroides*, and RpoE10 from *Azospirillum brasilense*, hold a C-terminal extension with a SnoaL-like domain. Contrary to the CorE-like ECF sigma factors, the C-terminal extensions of these proteins play an inhibitory role by contacting a motif in the linker region, preventing its binding to the RNAP core enzyme [31,53,56]. Part of the C-terminal extension, however, seems to be required for the proper conformation of the active ECF sigma factor, suggesting an activation mechanism mediated by a conformational change rather than by proteolytic cleavage of the C-terminal extension [25,53,54,57]. Other ECF sigma factor groups—such as ECF56, ECF205, ECF294, and ECF295—contain C-terminal extensions of 120–150 residues bearing SnoaL-like domains, although their regulatory functions remain to be elucidated [7].

#### 3.5.2. Activated by Protein Interaction

Like ECF sigma factors described in the previous section, these proteins are expected to undergo a conformational change mediated by their C-terminal extensions. However, unlike members of ECF238 and ECF41, this conformational change is not predicted to be prompted by the direct binding of the ligand to their C-terminal extensions, but by the interaction with other proteins.

This interaction-dependent regulation has been proposed for members of the group ECF42. Among them, the best studied are Sven_0747, Sven_7131, and Sven_4377, from *Streptomyces venezuelae* [58], and ECF-10 from *Pseudomonas putida*, which confers sensitivity to oxidative stress and antibiotics, and is homologous to ECF sigma factors of near 4000 other *Pseudomonas* strains [23,59]. These proteins hold a C-terminal extension of approximately 200 residues rich in tetratricopeptide repeat (TPR) domains, usually involved in protein–protein interactions, which is completely essential for the activity of these proteins [6,31,58]. As mentioned above, it is predicted that the interaction of the C-terminal extension with another protein results in a conformational change in the ECF sigma factor that activates/inactivates it (Figure 3B). Only the proximal region of the C-terminal extension is predicted to interact with the σ4 domain of the sigma factor [31], whereas the interaction partner(s) for the remaining extent of the protein remains to be identified. Nevertheless, the conserved presence of a YCII-related domain containing protein in their genetic neighborhood, and the fact that these domains can also be found fused to other ECF sigma factors (like in the case Q9A8M4_CAUVC from *Caulobacter vibrioides*), suggesting that this conserved protein is also involved in their regulatory mechanism.

A similar mechanism of regulation has been suggested for some members of the group ECF57 of sigma factors, which exhibits a prevalence of WD40-like repeats, typically involved in protein–protein interactions, in their C-terminal extensions [6]. The presence of two conserved cysteine residues that potentially link the σ2 and the σ4 domains suggest that there is a strong conformational change involved in the activation of these proteins. This link between both domains has been suggested to happen through a disulphide bridge between both cysteines [7]. However, due to the strong reducing conditions of the cytoplasm, other factors such as metal ligands are more likely to be involved in this process.

#### 3.5.3. Activated by Proteolysis

Proteins following this regulatory mechanism hold a N- or C-terminal extension that resembles anti-sigma factors, and inactivates the sigma factor (either membrane-bound or soluble). Thus, through an activation mechanism similar to that of the canonical ECF sigma factors activated by the RIP of their anti-sigma factors and the further proteolytic processing of its cytoplasmic portion, these regulatory extensions need to be processed by proteases to release an active transcriptional subunit.

To date, the only ECF sigma factor where this activation mechanism has been experimentally characterized is IutY from *P. putida* (ECF243). This protein is part of an iron-starvation cell surface signaling system, where the membrane-bound anti-sigma factor is fused to the C-terminal portion of the sigma factor. IutY responds to the xenosiderophore aerobactin to compete for the iron resources in the presence of other competitors. Upon activation of the system, the ECF sigma factor is subjected to RIP by the carboxypeptidase Prc and the transmembrane metalloprotease RseP to release the cytoplasmic portion of the sigma factor (Figure 3C). A truncated version of IutY, where most of the C-terminal extension was removed, resulted in an active protein, regardless of the presence of aerobactin, corroborating that the proteolytic processing of the sigma factor is enough to activate the protein [23,35].

Members of the ECF48, ECF52, and ECF53 groups of sigma factors from Actinobacteria are also expected to be regulated by this mechanism. Their domain structures suggest a C-terminal fusion with transmembrane portions functioning as anti-sigma factors, which hold regulatory extensions of up to 400 residues, comprising a zinc-finger domain (likely a typical ZAS domain), and from one to two transmembrane helices. Among them, the best understood is SCO4117 from *S. coelicolor* (ECF52), involved in antibiotic production, differentiation, and sporulation. This sigma factor has a periplasmic proline-rich region followed by a carbohydrate-binding domain. A truncated version of the protein demonstrated a significantly lesser effect than the deletion of the whole gene, supporting a model where the proteolytic processing of the C-terminal extension is activating the protein [34]. Phosphorylation of several residues on the sigma factor domain has also been observed for this protein, but its role in modulating the activity of SCO4117 is still not well understood [60].

Similarly, some members of the ECF36 group of sigma factors are also fused to their anti-sigma factors. They hold N-terminal extensions with three to four transmembrane helices and a DUF2275 domain. Other groups with regulatory extensions predicted to be regulated by proteolysis would include group ECF115, where their C-terminal extensions of approximately 70 residues are predicted to be degraded by Clp proteases, usually involved in the proteolysis of the cytosolic portion of anti-sigma factors, which are encoded in their genetic neighborhoods [7].

## 4. Outlook and Future Perspectives

Here we provided an updated view to the state of the art of the diverse mechanisms regulating the activity of non-canonical ECF sigma factors. Albeit, in recent years, many new pieces of evidence have been found, we are still far from completing this puzzle. Even though the most recent classification of ECF sigma factors described 157 groups, most of them lack a representative with experimental data, hindering the efforts to find and define the diverse activation mechanisms and biological roles of these regulatory proteins. Whereas the scientific community has made extraordinary steps in the discovery of new groups of ECF sigma factors, it seems that the next steps should focus on the characterization of members of the more than 100 groups that currently need experimental data to define their signature features. Moreover, further characterization is still required for most of the studied ECF sigma factors to get a good understanding of their activation mechanisms. The definition of these signature features is crucial to exploit these regulators for their roles in pathogenesis, to control the bacterial functions, to generate new biotechnological tools, and as targets for the development of new drugs.

## Figures and Tables

**Figure 1 ijms-23-03601-f001:**
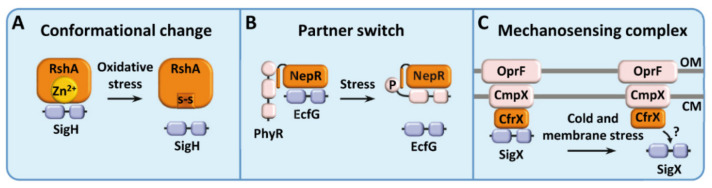
Regulation of non-canonical ECF sigma factors associated to soluble anti-sigma factors, exemplified by known ECFs and activating stresses. ECF sigma factors are depicted in blue, whereas proteins acting as anti-sigma factors and other regulatory proteins are depicted in orange and pink, respectively. (**A**) Regulation by conformational change: RshA and SigH from *M. tuberculosis*; (**B**) Partner switch in NepR-PhyR and EcfG complex from *R. sphaeroides*; (**C**) Model for the mechanosensing complex governing the response of SigX in *P. aeruginosa*. OM: outer membrane; CM: cytoplasmic membrane.

**Figure 2 ijms-23-03601-f002:**
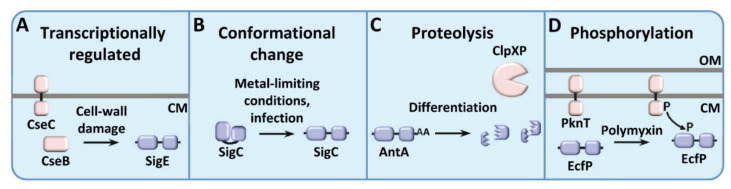
Examples of described regulatory mechanisms in ECF sigma factors non-associated to an anti-sigma factor. ECF sigma factors are depicted in blue, whereas additional regulatory proteins are depicted in pink. (**A**) SigE from *S. coelicolor* is transcriptionally regulated by the two-component system CseBC, which responds to cell wall damage; (**B**) SigC, involved in virulence of *M. tuberculosis*, is regulated by a conformational change under infection conditions; (**C**) AntA from *S. albus* is degraded by ClpXP during certain stages of the life cycle; (**D**) EcfP from *V. parahaemolyticus* is activated by phosphorylation by the STPK PknT in response to the antibiotic polymyxin. OM: outer membrane; CM: cytoplasmic membrane.

**Figure 3 ijms-23-03601-f003:**
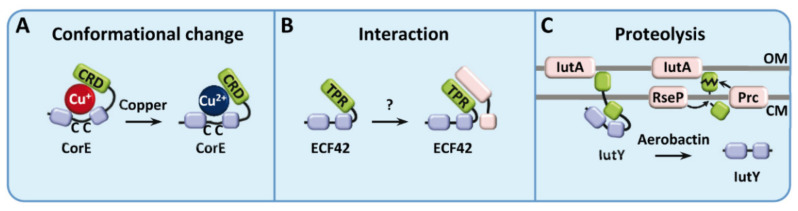
Anti-sigma factors governed by regulatory extensions. ECF sigma factors are depicted in blue, whereas their regulatory extensions and other regulatory proteins are depicted in green and pink, respectively. (**A**) The CorE C-terminal extension (CRD) from *M. xanthus* responds to copper redox state; (**B**) Regulation by protein interaction of ECF42; (**C**) IutY from *P. putida* is degraded by the carboxypeptidase Prc and the metalloprotease RseP. OM: outer membrane; CM: cytoplasmic membrane.

**Table 1 ijms-23-03601-t001:** Different regulatory mechanisms of the ECF sigma factors with soluble anti-sigma factors. The ECF groups are those that appear in the new classification of Casas-Pastor et al., 2021 [7]. * Only certain members of the group.

Proposed Mechanism	ECF Group	Model Sigma Factor	Reference
Conformational change	ECF11	RpoE from *Rhodobacter sphaeroides*	[9]
	ECF12	SigH from *Mycobacterium tuberculosis*	[10]
	ECF14	SigE from *Mycobacterium tuberculosis*	[11]
	ECF19 *	WP_016472479.1 from *Streptomyces albus*	[7]
	ECF293 *	RpoE from *Neisseria meningitidis*	[12]
Partner switch	ECF15	EcfG from *Methylobacterium extorquens*	[13]
Mechanosensing	ECF102	SigX from *Pseudomonas aeruginosa*	[14]
Unknown mechanism	ECF125	WP_044516075.1 from *Mycolicibacterium septicum*	[7]
	ECF127	EJO88542.1 from *Mycobacterium colombiense*	[7]
	ECF270	ODS58609.1 from *Acidobacteria bacterium* SCN 69–37	[7]
	ECF271	OGO36537.1 from *Chloroflexi bacterium* RBG_16_56_8	[7]
	ECF286	WP_003983642.1 from *Streptomyces rimosus*	[7]
	ECF292	WP_036395736.1 from *Mycolicibacterium cosmeticum*	[7]

**Table 2 ijms-23-03601-t002:** Regulatory mechanisms of the ECF sigma factors without anti-sigma factor. The ECF groups are those that appear in the new classification by Casas-Pastor et al., 2021 [7]. * Only certain members of the group.

Proposed Mechanism		ECF Group	Model Sigma Factor	References
Transcriptional regulation		ECF12 *	ECF12s9 and ECF12s2 from *Mycobacterium* sp.	[7]
		ECF32	HrpL from *Pseudomonas syringae*	[26]
		ECF39 *	SigE from *Streptomyces coelicolor*	[27]
		ECF114	SigH from *Porphyromonas* *gingivalis*	[28]
		ECF203	SCD72908.1 from *Streptomyces* sp. DvalAA-19	[7]
		ECF234	APQ59451.1 from *Paenibacillus polymyxa*	[7]
		ECF293 *	PA3285 from *Pseudomonas aeruginosa*	[14]
Conformational changes		ECF36 *	SigC from *Mycobacterium tuberculosis*	[7]
Proteolysis		ECF54	SFT86700.1 from *Geodermatophilus amargosae*	[7]
		ECF282	AntA from *Streptomyces albus*	[29]
Phosphorylation		ECF43	EcfP from *Vibrio parahaemolyticus*	[30]
		ECF59	SFI47409.1 from *Planctomicrobium piriforme*	[7]
		ECF61	OJW24604.1 from *Planctomycetales bacterium* 71–10	[7]
		ECF62	WP_008685225.1 from *Rhodopirellula sallentina*	[7]
		ECF217	ELP31162.1 from *Rhodopirellula baltica*	[7]
		ECF283	WP_056749340.1 from *Nocardioides* sp. Root190	[7]
Withregulatory extensions	Conformational change	ECF41	SigJ from *Mycobacterium tuberculosis*	[31]
		ECF238	CorE from *Myxococcus xanthus*	[32,33]
	Protein interaction	ECF42	Sven_0747 from *Streptomyces venezuelae*	[31]
		ECF57 *	WP_015250107.1 from *Singulisphaera acidiphila*	[6]
	Proteolysis	ECF36 *	KLO31890.1 from *Mycolicibacter heraklionensis*	[7]
		ECF48	WP_048473130.1 from *Mycolicibacterium chlorophenolicum*	[7]
		ECF52	SCO4117 from *Streptomyces coelicolor*	[34]
		ECF53	WP_030276194.1 from *Streptomyces purpeochromogenes*	[7]
		ECF115	KOP67510.1 from *Bacillus* sp. FJAT-18019	[7]
		ECF243 *	IutY from *Pseudomonas putida*	[35]
		ECF270 *	WP_011419852.1 from *Anaeromyxobacter dehalogenans*	[7]
	Others	ECF29	SED43577.1 from *Bradyrhizobium lablabi*	[7]
		ECF56	WP_042440600.1 from *Streptacidiphilus albus*	[7]
		ECF123 *	WP_028426757.1 from *Streptomyces* sp. TAA040	[7]
		ECF205	WP_019068201.1 from *Streptomyces hokutonensis*	[7]
		ECF216 *	QDE78790.1 from *Myxococcus xanthus*	[7]
		ECF220	WP_061622786.1 from *Sorangium cellulosum*	[7]
		ECF237	OLT65459.1 from *Moorea producens*	[7]
		ECF240 *	SIO28919.1 from *Chryseobacterium scophthalmum*	[7]
		ECF262	SFB89493.1 from *Ruminococcus albus*	[7]
		ECF264	WP_037286607.1 from *Saccharibacillus sacchari*	[7]
		ECF276 *	WP_063815919.1 from *Sorangium cellulosum*	[7]
		ECF287	WP_033089221.1 from *Nocardia seriolae*	[7]
		ECF288	WP_018594055.1 from *Blautia producta*	[7]
		ECF294	AKZ62584.1 from *Herbaspirillum hiltneri*	[7]
		ECF295	WP_063065904.1 *Nocardia violaceofusca*	[7]
Unknown mechanism		ECF58	APZ92118.1 from *Fuerstia marisgermanicae*	[7]
		ECF122	WP_057211282.1 from *Cellulomonas* sp. Root930	[7]
		ECF201	CDO03659.1 from *Oceanobacillus picturae*	[7]
		ECF248	EOZ99538.1 from *Indibacter alkaliphilus*	[7]
		ECF257	WP_010287217.1 from *Kurthia massiliensis*	[7]
		ECF265 *	AKO94994.1 from *Bacillus endophyticus*	[7]

## Data Availability

Not applicable.

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
