# Peer review of "Mechanisms of Action of Non-Canonical ECF Sigma Factors"

_ijms, 2022, doi:10.3390/ijms23073601_

Round 1

Reviewer 1 Report

The manuscript of the review article summarizes the knowledge of mechanisms by which so-called non-canonical ECF sigma factors are regulated. The manuscript is well written, is comprehensive and clearly describes the state of art in this field. Some minor corrections are necessary to improve it. However, I have also some comments to the general strategy how to explain the knowledge with many facts (major comment 4).

Major comments:

1) The term “ECF sigma factors” is somewhat loosely defined. I suggest to provide information about its first use and precise meaning. The group was in fact named Group 4 by Gruber and Gross. Are these groups really identical? The Group 4 is defined by the number of subdomains, how ECF is defined? Are all members of the group involved in stress responses? I suggest to include: Group 4 (Gruber and Gross, 2003) also called ECF…

2) I have the same concern for the term “canonical” and “non-canonical”. It seems that this word is misused in the last years. Please, explain why some group of sigma factors is called canonical and what does it mean. Maybe, these factors were just described earlier.

3) It should be mentioned that many defined ECF groups according to Staron et al. contain only a single sigma member. This makes the sigma grouping problematic and shows a wide variety of sigma types. Please, make some comment to this.

4) At many places in the text, sigma factors and mechanism are only enumerated with not much detail. The methods by which the knowledge was found out are not mentioned, but might be important for reads. I suggest to add some paragraphs, which would make the review more interesting: Choose (in some groups) one example of a sigma and mechanism, which are very interesting, surprising or difficult to study and describe shortly history of its investigation. Include methods, which were used to explain their mechanism of action. Such “story” would make the text more lively and attractive.

Minor comments:

  • Please, do not mix up “sigma factor” and “mechanism”. E.g. page 2, line 73-74: Among the ECF sigma factors…..we find three well-defined mechanism.

Expression “mechanism...of sigma factor” sounds me strange. There might be Mechanism of sigma factor inhibition, mechanism of sigma factor activity, mechanism of sigma factor activation. Maybe "mechanism of sigma factor regulation" is better than  "regulatory mechanism of sigma factor"

  • 3, line 94: Examples of this mechanism would include RpoE…. Mycobacterium tuberculosis. Please, provide reference.
  • Many Latin names of bacterial species are not in italics. Please, check and correct. Page 4, Line 13,page 6, line 22, page 7, lines 23, 24,25, 26 and other places
  • Page 3, line 11: . binding to Zn2+ and to the ECF… - delete “and?”
  • Page 6, line 18: plant infection of this pathogen… probably: plant infection by this pathogen..
  • Page 7, line 24: Caspase HetF Associated … I believe “caspase HetF associated…” is better
  • Page 7, line 26: which contains negatively charged residues…, interact with positively charged…, the same page 8, line 28: negatively instead of negative

Author Response

I am enclosing a pdf with the point-by-point response to comments to reviewer 1

Reviewer 2 Report

The review of Marcos-Torres is clear and well written. The authors make a comprehensive view of different activation mechanisms of non-canonical extracytoplasmic function sigma factors (ECF) and for this reason their review is of relevance  to the field. Moreover, they provide an update on regulatory mechanisms of the non-canonical ECF sigma factors, taking in account several microorganisms of interest. The classification of non-canonical ECF and  their descriptions are well described. The bulk of cited references are appropriate and coherent with the text and their conclusions. What I would to improve in this review are the tables, inserting something more about the organisms in which the cited ECF was found and some special characteristic if someone non-completely falls in the general classification.

Author Response

I am enclosing a pdf file with point-by-point response to the reviewer 2 comments 
